# Effects of Pulsed Magneto-Oscillation on the Homogeneity of Low Carbon Alloy Steel Continuous Casting Round Billet

Yingchun Wang [1], Rongjun Xu [1], Honggang Zhong [2,*], Guodong Xu [1], Zhishuai Xu [2], Renxing Li [2] and Qijie Zhai [2]

1   Central Research Institute of China Baowu Iron and Steel Group Co., Ltd., Shanghai 201900, China; wangyingchun@baosteel.com (Y.W.); xurongjun@baosteel.com (R.X.); xugd@baosteel.com (G.X.)
2   Center for Advanced Solidification Technology, Shanghai University, Shanghai 200444, China; xzsshu@163.com (Z.X.); renxingli@shu.edu.cn (R.L.); qjzhai@shu.edu.cn (Q.Z.)
*   Correspondence: hgzhong@shu.edu.cn

**Abstract:** Pulsed Magneto-Oscillation (PMO) is a newly developed and effective homogenization technique, and has been successfully applied in rectangular continuous casting, but its processing parameters and effective stability in round billet continuous casting have not been investigated. In this paper, the effects of PMO on the solidification structure and the macrosegregation of Φ 178 mm continuous casting round billets for low carbon alloy steel were studied by industrial experiments. The results show that PMO can stably increase the equiaxed grain area, and reduce the macrosegregation of billets. Moreover, it has strong adaptability to steel grade and continuous casting process parameters. Compared with the billets without PMO treatment, for 93.8% of billets (15 billets) solidified with PMO, the equiaxed grain area ratio increased by an average of 5.8%, while for 87.5% of billets (14 billets), the carbon segregation index range decreased by an average of 0.06, though different steel grades, superheat and casting speed were used in the experiment. It is believed that convection caused by Lorentz force can accelerate the heat dissipation of steel liquid, and reduce the temperature of a liquid at the solidification front, while the magnetic oscillation effect is conducive to dendrite fragmentation. Both effects lead to refinement of the solidification structure and reduction of macrosegregation.

**Keywords:** PMO; homogenization; continuous casting; round billet; low alloy steel

## 1. Introduction

Continuous casting is widely used by metallurgical enterprises, for more than 90% of raw steel in the world. Serious macrosegregation and central shrinkage defects are always formed in the continuous casting billets, due to forced cooling and large liquid core during continuous casting, which only partially eliminated by subsequent hot working and reduction of the billet quality [1].

Homogenization technologies are usually adopted to reduce macrosegregation or to refine the solidification structure in order to improve the billet quality, including low superheat pouring, soft/heavy reduction [2], electromagnetic stirring [3] and permanent magnet stirring [4]. Pouring steel liquid with 10 °C superheat or below is quite difficult owing to the limitations of inclusions removal and the freezing risk of molten steel in the ladle, and high superheat is unfavorable for obtaining fine equiaxed grain structures. Soft/heavy reduction technology could reduce shrinkage cavity and central segregation by inducing deformation at the final solidification stage of the billets. However, accurate prediction and precise control of the liquid core are necessary, which is a great challenge for the continuous casters producing different kinds of steels [5]. Despite the wide application of electromagnetic stirring in continuous casting, homogenization is only achieved by vigorous stirring, and serious negative segregation band (i.e., white band) occurs with improper control [6].

In recent years, refining the solidification structure by physical fields has developed rapidly. Ultrasonic [7], pulsed current [8] and pulsed magnetic field [9] are used to refine grain structures [10]. Gong and Zhai, et al., [11] put forward a Pulsed Magneto-Oscillation (PMO) technique, based on the grain refinement mechanism of electric current pulse, to gain the desired microstructure [12,13], which has been applied to produce homogeneous rectangular continuous casting billets [14–16]. Gong [11,17] found that PMO can promote heterogeneous nucleation, and then the nuclei under the influence of electromagnetic force and liquid flow to form grain rain, and move to the liquid core and bottom, thus refining the solidification structure and improving the uniformity. Edry [18] believes that fragmented dendrites caused by Lorentz force, and cavitation induced by PMO, contribute to the refinement of the solidification structure. St. John [10] agrees with the shedding mechanism, which considers the oscillation effect of shedding the grains formed on the mold wall or liquid surface to increase nucleation, instead of the cavitation effect caused by PMO and electric current pulse [11,12]. These studies are valuable for understanding the solidification structure refinement in ingots treated by PMO. For continuous casting billets, however, the effective treatment time of the magnetic field is much shorter than ingots. Consequently, further mechanism discussion is needed about the effects of PMO on the solidification structure of continuous casting billets. Meanwhile, for the industrial application of the PMO technique, its influence on the solidification homogenization of small sections of round billet, and its stability, are also worth studying.

In this paper, the effects of PMO on the solidification structure and macrosegregation of Φ 178 mm continuous casting round billets of five grades of low carbon alloy steels were studied by industrial experiments. The mechanism of improving billets homogenization by PMO and the influence of process parameters were analyzed.

## 2. Experimental Process

### 2.1. Experimental Materials and Methods

The materials used in the industrial experiments are listed in Table 1. There were five grades of steel, marked as S1–S5 in turn, and a total of sixteen experiments were carried out.

**Table 1.** Experimental steels and the alloy composition (wt.%).

| Materials | Number | C | Mn | Si | Cr | Mo | V | Fe |
|---|---|---|---|---|---|---|---|---|
| S1 | 1 | 0.09 | 1.25 | 0.27 | 0.07 | 0.082 | 0.038 | Balance |
|  | 2 | 0.09 | 1.24 | 0.26 | 0.04 | 0.081 | 0.038 | Balance |
|  | 3 | 0.09 | 1.26 | 0.25 | 0.04 | 0.083 | 0.038 | Balance |
| S2 | 4 | 0.26 | 0.80 | 0.27 | 1.02 | 0.215 | 0.006 | Balance |
|  | 5 | 0.26 | 0.78 | 0.28 | 1.03 | 0.230 | 0.006 | Balance |
| S3 | 6 | 0.25 | 0.98 | 0.24 | 1.07 | 0.013 | 0.005 | Balance |
|  | 7 | 0.25 | 1.02 | 0.28 | 1.07 | 0.007 | 0.005 | Balance |
|  | 8 | 0.25 | 1.00 | 0.23 | 1.06 | 0.012 | 0.005 | Balance |
| S4 | 9 | 0.37 | 1.25 | 0.24 | 0.04 | 0.008 | 0.032 | Balance |
|  | 10 | 0.37 | 1.28 | 0.24 | 0.05 | 0.018 | 0.034 | Balance |
|  | 11 | 0.36 | 1.23 | 0.18 | 0.04 | 0.007 | 0.037 | Balance |
|  | 12 | 0.35 | 1.25 | 0.22 | 0.05 | 0.011 | 0.030 | Balance |
| S5 | 13 | 0.26 | 0.51 | 0.21 | 0.49 | 0.731 | 0.101 | Balance |
|  | 14 | 0.25 | 0.47 | 0.24 | 0.49 | 0.729 | 0.103 | Balance |
|  | 15 | 0.27 | 0.49 | 0.31 | 0.49 | 0.750 | 0.100 | Balance |
|  | 16 | 0.26 | 0.47 | 0.27 | 0.48 | 0.762 | 0.104 | Balance |

A six-strand full arc round billet continuous caster was chosen for PMO industrial experiments, in which the 5th strand was contrast samples without PMO and the 6th strand was treated by PMO. The cross-sectional size of the billets is Φ 178 mm. The schematic of the continuous caster and the position of the induction coil of PMO is shown

in Figure 1. The PMO coil was located under the mold and foot roller, the height was 0.33 m, and the center of the coil was 1.3 m away from the meniscus. The main caster parameters, including mold electromagnetic stirring (M-EMS) and final electromagnetic stirring (F-EMS), are shown in Table 2. Table 3 shows the main casting parameters during the industrial experiments. The range of superheat in the tundish was 27–44 °C, the strand casting speeds were 1.80–2.50 m/min, the impulse frequency of the PMO were 250–700 Hz, and the PMO power range was 65–105 kW.

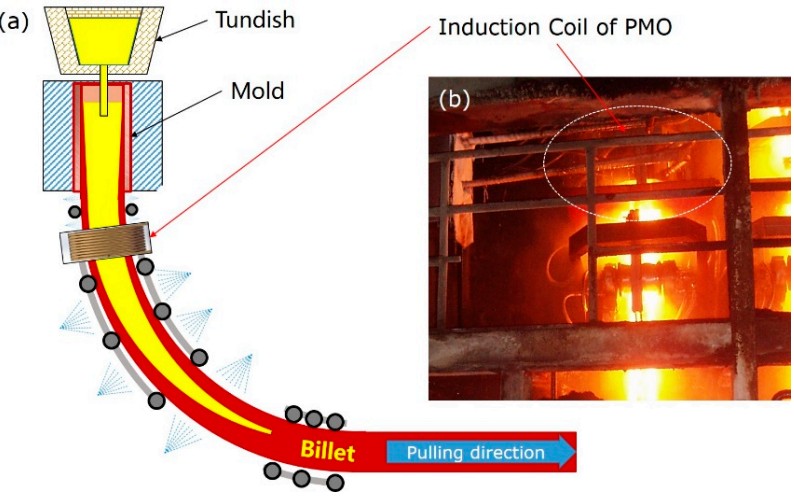

**Figure 1.** Schematic of the continuous caster showing (**a**) the position of the induction coil and (**b**) photo of the induction coil of PMO on the billet.

**Table 2.** Main parameters of the continuous casting machine.

| Parameters | Value |
|---|---|
| Effective length of mold | 800 (mm) |
| Billet size | $\Phi$ 178 (mm) |
| Arc radius | 10 (m) |
| Cooling mode of secondary cooling zone | Aerosol cooling |
| M-EMS electric current | 250 (A) |
| F-EMSelectric current | 350 (A) |

**Table 3.** Experimental parameters.

| Cast No. | 1 | 2 | 3 | 4 | 5 | 6 | 7 | 8 | 9 | 10 | 11 | 12 | 13 | 14 | 15 | 16 |
|---|---|---|---|---|---|---|---|---|---|---|---|---|---|---|---|---|
| Superheat (°C) | 40 | 39 | 40 | 29 | 28 | 44 | 33 | 33 | 28 | 35 | 42 | 40 | 41 | 33 | 29 | 27 |
| Casting speed (m/min) | 1.95 | 1.9 | 1.95 | 2.1 | 2.4 | 1.8 | 2.11 | 2.5 | 2.5 | 2.4 | 2.1 | 2.1 | 1.8 | 2.05 | 1.9 | 1.9 |
| Impulse frequency of PMO (Hz) | 600 | 700 | 500 | 250 | 300 | 350 | 400 | 300 | 250 | 300 | 350 | 400 | 600 | 700 | 250 | 300 |
| Power of PMO (kW) | 75 | 85 | 105 | 65 | 75 | 70 | 80 | 90 | 65 | 80 | 85 | 90 | 65 | 70 | 85 | 95 |

## 2.2. Sampling and Testing Methods

All billets treated with or without PMO were taken during the middle period of pouring (1/2 molten steel remaining in ladle), and the corresponding superheat and casting speed were recorded. The slice samples with the thickness of 30 mm were sawed from the billets, and milled to surface roughness (Ra) less than or equal to 1.6 μm. Then, the samples were etched at 70 °C for 30 min by 50% HCl water solution. The macrostructure was photographed with a camera.

Along the center line, from the surface to the center of the slice samples, two samples of 20 × 45 × 10 mm and 20 × 48 × 10 mm were cut to observe the solidification structure, as shown in Figure 2a. After grinding and polishing, the samples were etched in supersaturated trinitrophenol aqueous solution at 75 °C, and then the microstructure was observed and photographed by optical microscope.

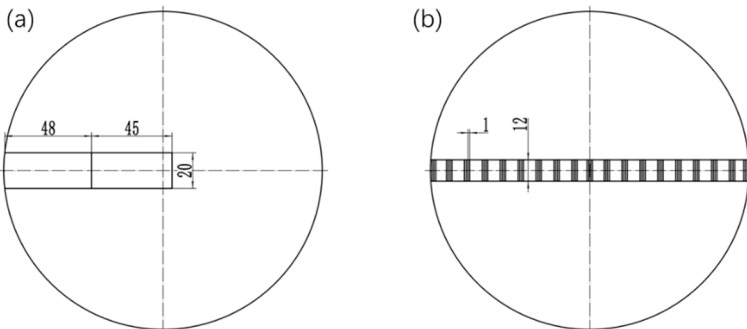

**Figure 2.** Sampling position of (**a**) microstructure and (**b**) carbon content on cast billet.

To detect carbon distribution along the radial direction of the billets, 19 points with 9.6 mm intervals were selected on each billet. Three specimens, $12 \times 1 \times 6$ mm in size, were cut by wire-electrode cutting at each testing point (Figure 2b), and the carbon contents of the specimens were detected by infrared carbon and sulfur analyzer (CS2800, Steel Inak, Beijing, China). The mean value of the three specimens was taken as the carbon content at this point. The carbon segregation index ($K$) was used to characterize the degree of carbon segregation, and the calculation formula was as follows:

$$K = C_i / \overline{C} \tag{1}$$

where $C_i$ is the carbon content of each detection point, and $\overline{C}$ is the arithmetic average of the carbon content of 19 detection points.

## 3. Results

### 3.1. Solidification Structure and Macrosegregation

Taking the No. 11 billets as an example, the effects of PMO on the solidification structure and macrosegregation are discussed. The macrostructures of the No. 11 billets with and without PMO are shown in Figure 3. For the billet without PMO (as shown in Figure 3a), there was a 3.2 mm shrinkage cavity in the center, while for the billet with PMO (as shown in Figure 3b), there was no shrinkage cavity in the center. Furthermore, comparing the billet treated with PMO to the billet without PMO, the equiaxed grain area ratio was much lower, and the porosities in the central equiaxed grain zone were much higher.

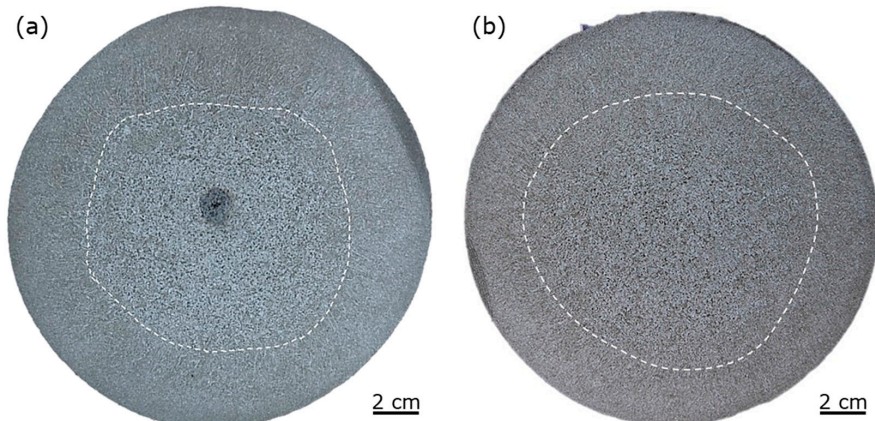

**Figure 3.** Macrostructure on cross section of low-alloy steel S4 (No. 11) round billet (**a**) without PMO and (**b**) with PMO.

Figure 4 shows the microstructure of the No.11 billets cast with and without PMO respectively. Both billets contained chill zone, columnar grain zone, coarse dendritic grain zone, and equiaxed grain zone. As shown in Figure 4, PMO made a great contribution to

the reduction of the coarse dendritic grain zone and the lengths of primary dendritic arms, which was consistent with the results of the macrostructure analysis. By PMO-treating, the area ratio of the equiaxed grain zone in the billet center increased, and the dendrites in front of the columnar grains were obviously refined.

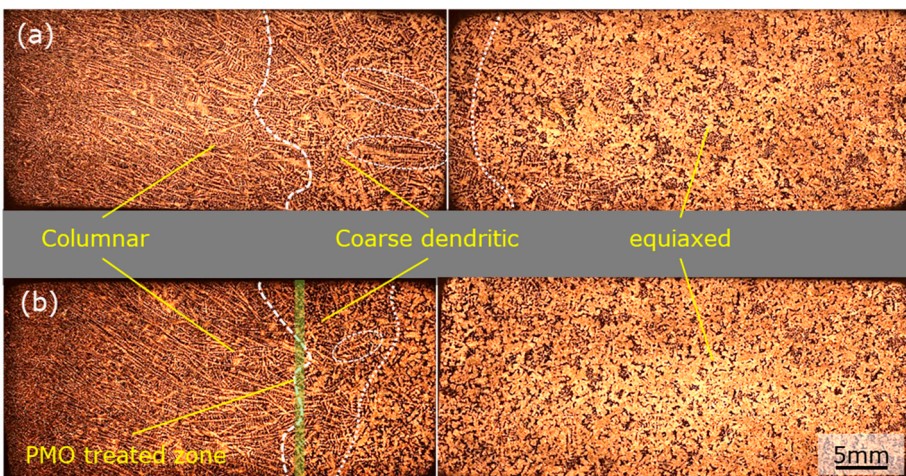

**Figure 4.** Microstructure in radial direction of No. 11 billet sample (**a**) without PMO and (**b**) with PMO.

The radial distribution of the carbon segregation index in the cross section of the No.11 billets is shown in Figure 5. Compared with the billets without PMO treatment, the homogeneity of the carbon distribution in the billets with PMO was much better, and the positive segregation at 60 mm to the center was smaller (usually called 1/4 diameter segregation). The improvement of carbon distribution homogeneity was related to the reduction of the dendrites size in front of the columnar zone, where the solidification front was when the billet passed through the induction coil of PMO.

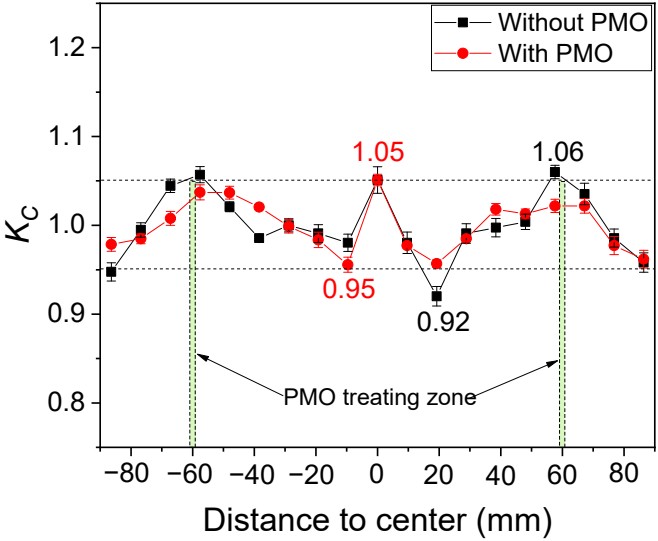

**Figure 5.** Macrosegregation in radial direction of No.11 billet sample.

### 3.2. Statistics of Central Equiaxed Grain Area Ratio and Carbon Segregation Range

Figures 6 and 7 show the statistics of the central equiaxed grain area ratio and carbon segregation range along the radial direction of 16 experimental billets. As shown in Figure 6, compared to the billets without PMO treatment, in the 15 experimental billets with PMO treatment, the equiaxed grain area ratio increased by an average of 5.8%, and in the 9 billets with PMO treatment, the equiaxed grain area increased by more than 20%.

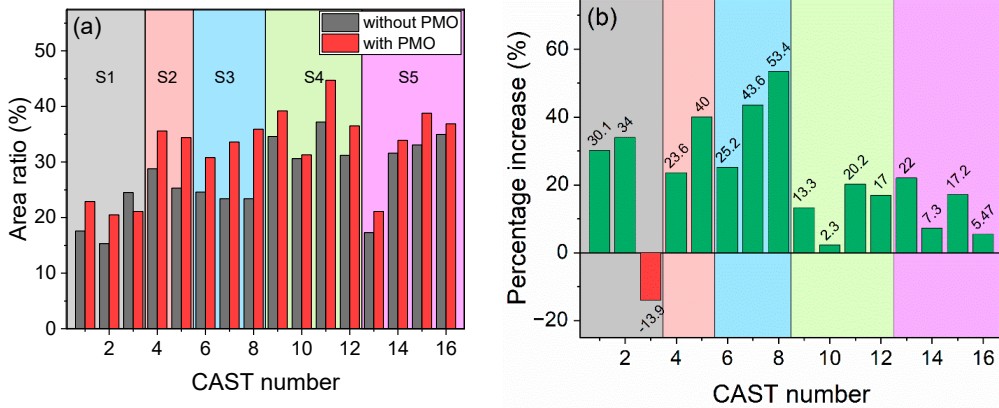

**Figure 6.** (**a**) The area ratio of the equiaxed grain zone of the experimental billets and (**b**) the percentage increase in equiaxed grain area of billet with PMO treatment.

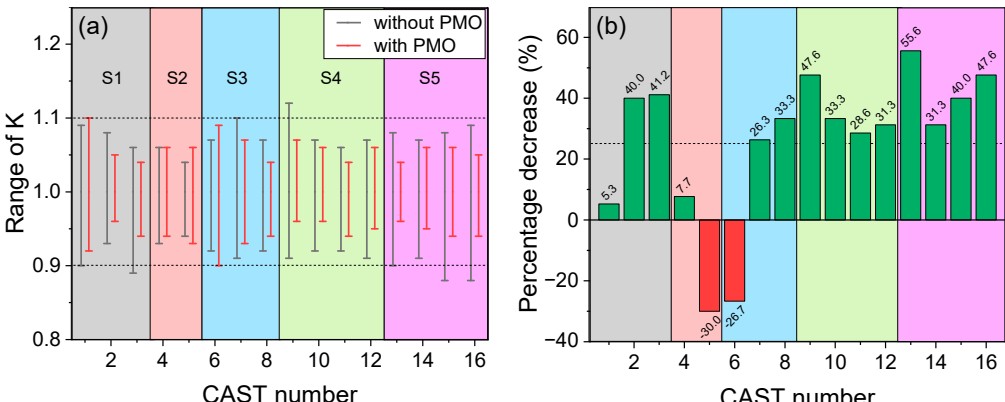

**Figure 7.** The $K_C$ range of the billet (**a**) and the percentage decrease (**b**) in the $K_C$ range of the billet with PMO to without PMO.

As shown in Figure 7a, among the 16 PMO treated billets, the carbon segregation index range of 14 billets decreased by an average of 0.06, and for 75% of the 14 billets, the carbon segregation index range decreased by more than 25% (Figure 7b). This shows that PMO is widely adaptable for continuous casting processes and steel grades. In addition, as shown in Figure 7a, most of the carbon segregation index of the billets without PMO treatment was between 0.9 and 1.1, indicating that the macrosegregation was well controlled, and that PMO still had an homogenization effect on almost all the billets.

## 4. Discussion

### 4.1. Mechanism Analysis

PMO evidently could increase the equiaxed grain ratio in the center of the round billets, refine the dendrites, and reduce the carbon segregation. In accordance with the production conditions, the time for a certain billet section across the PMO coil was quite short, on account of the continuous movement of the billet along the casting direction. Considering the casting speeds selected in the experiment, the time was only 7.9 s to 11.0 s. The solidification process of continuous casting billets in a secondary cooling zone differs from ingots in that there is no free liquid surface and mold wall for nucleation. Consequently, the mechanism of refining the solidification structure of continuous casting billets by PMO should be further discussed.

Given the characteristics of electromagnetic induction, the solenoid coil used for the PMO could form two circulations [17,19] in the liquid phase of the billet. As the billets moved, the influence zone of the upper circulation was smaller than that of the lower circulation (as shown in Figure 8). Compared to the billets without PMO treatment, the

double circulations enhanced the heat transfer between the liquid phase and the solidified shell, accelerated the heat dissipation of the liquid phase, and changed the temperature distribution in the liquid phase of the billets. As a result, the temperature of the liquid ($T_1$ in Figure 6) in front of the freezing dendrites in the upper circulation zone increased, while that in front of the dendrites in the lower circulation zone decreased. This temperature distribution will cause the growth rate of the solidified shell to decrease first and then increase, as shown in the enlarged part of Figure 8. The temperature of the liquid in the lower circulation zone ($T_2$ in Figure 6) is lower than that in the upper circulation zone, or even supercooled, leading to a thermal undercooling at the freezing front, which is beneficial to the nucleation and growth of equiaxed grains. In addition, the forced convection caused by Lorentz force and the interdendritic melt vibration generated by PMO could also promote the dendrite remelting and forming more dendritic fragments. The microstructure of the coarse dendritic zone with PMO treatment in Figure 4b is obviously refined, which provides evidence that PMO promotes nucleation or fragment of dendrite. The combination effects of the grain refinement in this zone and the forced convection led by PMO could reduce the solute enrichment at the front of the dendrites, thus reducing the positive segregation (Figure 5) at 1/4 diameter. On the other hand, the grain nuclei and dendritic fragments formed along the supercooled zone in the freezing front, drifted to the center of the billet under the promotion of electromagnetic force and flow field, which is beneficial for increasing the number of central equiaxed grains and refining them [20]. Many studies have shown that microstructure refinement has an obvious effect on the improvement of macrosegregation, so the segregation in radial direction of billets treated by PMO could also be significantly reduced owing to the grain refinement in columnar to equiaxed transition zone and central equiaxed zone, and the homogenization of billets could be improved.

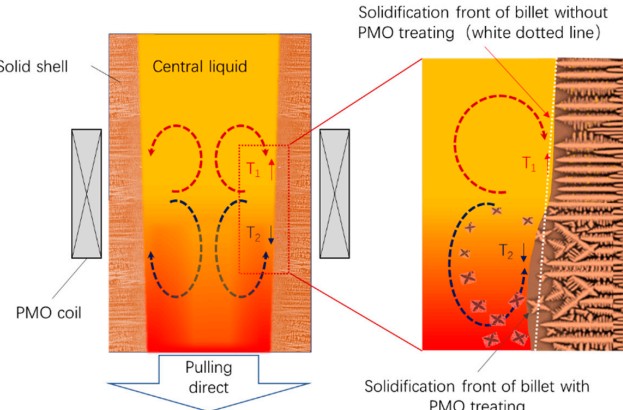

**Figure 8.** Schematic diagram of the effects of PMO on the temperature field of liquid phase and solidified shell growth of the billet (longitudinal cross section).

### 4.2. Statistics Analysis

According to the principles of heat transfer and solidification, high casting speed could reduce the effective treating time of PMO, thus decreasing the variation of temperature field, which hinders nucleation along the solidification front, and reduces the number of dendrite fragments. Consequently, high casting speed plays a negative role on the homogenization. In this study, the increase rate of equiaxed grain ratio for three kinds of steels had negative correlation with casting speed. For the S1-No. 3 billet, however, the equiaxed grain rate decreased, owing to the remarkable heating effect of high PMO power (105 kW) on the billet, which decreased the undercooling in the freezing front of the billet.

Figure 9 shows the effect of process parameters on the reduction of the carbon segregation index (*K*) range of different steel billets. Figure 9a shows that there is no clear relationship between the reduction rate of the *K*-range and casting parameters, and the improvement degree of carbon segregation in different steels varies under the similar superheat. The high

superheat and casting speed have negative effects on the decrease of carbon segregation. As shown in Figure 9b, there is a trough in the influence of the PMO parameters on the *K*-range, corresponding to low frequency and low power range, which mean that PMO should exceed a minimum value to achieve homogenization effect. Under the experimental conditions, high frequency and low power processing were conducive to obtaining a small *K*-range. Figure 10 shows the correlation between the solute content and the decrease of the *K*-range, respectively. The increase of V content seems beneficial to the decrease of the *K*-range under PMO treatment; it may be related to the oxide of V, but the trend was not significant enough to be conclusive. Meanwhile, the other elements had little effect.

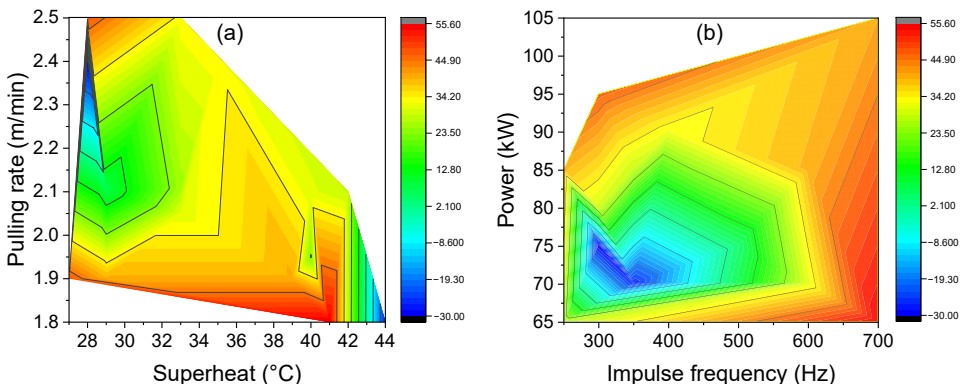

**Figure 9.** The decreasing percentage of carbon segregation (*K*) range in the radial direction of the billet via (**a**) the parameters of casting and (**b**) PMO.

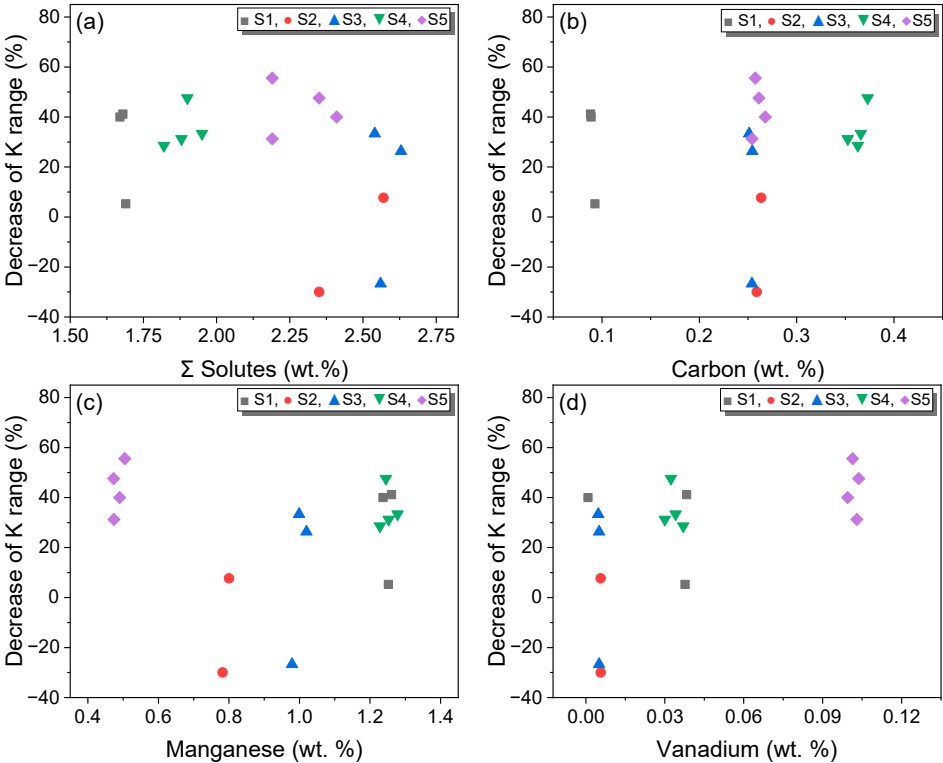

**Figure 10.** The decreasing percentage of carbon segregation (*K*) range in the radial direction of the billet via solute content: (**a**) total solute content; (**b**) C; (**c**) Mn; (**d**) V.

## 5. Conclusions

(1)     PMO can stably increases the equiaxed grain area, and reduces the macrosegregation of Φ 178 mm continuous casting round billets. Among the 16 experimental billets, the

(2) equiaxed grain area in 15 billets increased, and the carbon segregation in 14 billets improved. The superheat, casting speed, and main alloy element content had little correlation with the PMO treatment effect, which indicates that PMO can adapt to a wide range of casting parameters and different steels.

(2) The mechanism of the homogeneity of Φ 178 mm continuous casting round billets improved by PMO is that with the effects of magneto-oscillation, the heat dissipation of the liquid core accelerates, and the temperature of a liquid along the freezing front in the lower circulation zone reduces. Consequently, nuclei and dendrites fragmentation increases, which promotes grain refinement and segregation reduction.

**Author Contributions:** Conceptualization, Q.Z.; methodology, Q.Z. and R.L.; validation, H.Z. and Y.W.; investigation, R.L., Y.W. and H.Z.; resources, R.X., G.X. and Q.Z.; writing—original draft preparation, Y.W. and H.Z.; software, Z.X.; writing—review and editing, H.Z.; supervision, Q.Z.; funding acquisition, Q.Z. All authors have read and agreed to the published version of the manuscript.

**Funding:** This research was funded by the National Key Research and Development Program of China (No. 2020YFB2008400) and the National Natural Science Foundation of China (52130109).

**Institutional Review Board Statement:** Not applicable.

**Informed Consent Statement:** Not applicable.

**Data Availability Statement:** The data presented in this study are available upon request from the corresponding author.

**Conflicts of Interest:** The authors declare no conflict of interest.

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
