# Peer review of "Effects of Pulsed Magneto-Oscillation on the Homogeneity of Low Carbon Alloy Steel Continuous Casting Round Billet"

_metals, doi:10.3390/met12050833_

Round 1
Reviewer 1 Report
The manuscript under consideration is “Effects of pulsed magneto-oscillation on the homogeneity of low carbon alloy steel continuous casting round billet”. The subject of this research is practically important and scientifically valuable. From my point of view, the article deserves publishing after making some corrections.
- The scientific novelty of the results should be emphasized in abstract.
- In Table 1, kindly check the necessity of the third decimal points in carbon content values. The same question for other elements except Mo and V.
- Strand casting speed is given in [m], maybe [m/s]? (section 2.1., the second paragraph)
- Tables 1, 2, 3 are not referenced in the text.
- Kindly consider replacing “liquid temperature” to “temperature of a liquid” (or something like that) in abstract and throughout the text.
- In Table 2, kindly consider replacing “Project” to “Parameter”.
- “Error! Reference source not found.” often goes instead of references to literature or figures. Kindly correct.
- Figures 7, 9, 10 are mistakenly given as Figures 1, 2, 3.
- Kindly argue how you conclude there are correlations shown on Figures 10 a-d.
- Capture of Figure 6 does not mention variant (b). Also, I cannot understand the meaning of “to without”.
Author Response
Dear Reviewer,
Thank you for your kindly comments concerning our manuscript. Those comments are all valuable and very helpful. We have made revisions according to the comments. Revised portion are in red in the revised manuscript. We are outlining the major changes in this letter.
Responses to the comments
- The scientific novelty of the results should be emphasized in abstract.
Response: The scientific novelty of the results has been supplemented in the abstract.
- In Table 1, kindly check the necessity of the third decimal points in carbon content values. The same question for other elements except Mo and V.
Response: Thank you very much for your reminding. We rechecked the test method and accuracy. The content of each element in the paper is accurate to the secondary decimal points except Mo and V.
- Strand casting speed is given in [m], maybe [m/s]? (section 2.1., the second paragraph)
Response: The unite of casting speed is m/min, corresponding modifications have been made in the manuscript.
- Tables 1, 2, 3 are not referenced in the text.
Response: Corresponding modifications have been made in the manuscript.
- Kindly consider replacing “liquid temperature” to “temperature of a liquid” (or something like that) in abstract and throughout the text.
Response: Corresponding modifications have been made in the manuscript.
- In Table 2, kindly consider replacing “Project” to “Parameter”.
Response: Corresponding modifications have been made in the manuscript.
- “Error! Reference source not found.” often goes instead of references to literature or figures. Kindly correct.
Response: The Figures and Tables in the manuscript have been re-referenced.
- Figures 7, 9, 10 are mistakenly given as Figures 1, 2, 3.
Response: Corresponding modifications have been made in the manuscript.
- Kindly argue how you conclude there are correlations shown on Figures 10 a-d.
Response: The increase of V content seems beneficial to the decrease of K-range under PMO treatment, it may be related to the oxide of V, but the trend is not significant enough to be conclusive. And, we revised the statement in the manuscript.
- Capture of Figure 6 does not mention variant (b). Also, I cannot understand the meaning of “to without”.
Response: “with PMO to without” mean “applying PMO versus not applying PMO”. We changed the figure caption, and hope it has been stated more clearly.
Reviewer 2 Report
This work studies the effects of pulsed magneto-oscillation (PMO) on the solidification structure and the macrosegregation of continuous casting round billets for low carbon alloy steel through industrial experiments.
Although this work presents helpful information, the manuscript cannot be accepted in its current form. The main reasons for this recommendation are listed below.
General comments:
- The manuscript readability must be improved. Several sentences are miswritten, for example, the last sentence in the second paragraph of the Introduction section. This sentence could be written as "Despite the wide application of electromagnetic stirring in continuous casting, homogenization is only achieved by vigorous stirring, and serious negative segregation band (i.e., white band) occurs with improper control [6]."
- Several typos must be corrected, for example, units of the casting speed reported in the last line of page 2.
- In the manuscript, several cross-reference links to figures and tables are broken.
- The numbering of several figures is incorrect.
- References do not follow the MDPI style.
Technical comments:
Several results reported in this manuscript coincide with the findings of quite old works, such as that of Poppmeier et al., 1966.
Poppmeier, W., Tarmann, B., & Schaaber, O. (1966). Application of alternating electromagnetic fields in the continuous casting of steel. JOM Journal of the Minerals Metals and Materials Society, 18(10), 1109–1114.
Moreover, similar results were already reported in previous works of the authors' research group. Therefore, the authors must state precisely the contributions of this work to state-of-the-art.
The Conclusion and the Article Highlights sections are practically the same. Thus, the authors must unify these two sections in the Conclusions section.
Author Response
Dear Reviewer,
Thank you for your comments concerning our manuscript. Those comments are all valuable and very helpful. We have made revisions according to the comments. Revised portion are in red in the revised manuscript. We are outlining the major changes in this letter.
Responses to the comments:
General comments:
1. The manuscript readability must be improved. Several sentences are miswritten, for example, the last sentence in the second paragraph of the Introduction section. This sentence could be written as "Despite the wide application of electromagnetic stirring in continuous casting, homogenization is only achieved by vigorous stirring, and serious negative segregation band (i.e., white band) occurs with improper control [6]."
Response: Corresponding modifications have been made in the manuscript, and we also checked the whole manuscript and polished it.
2. Several typos must be corrected, for example, units of the casting speed reported in the last line of page 2.
Response: The units of the casting speed is m/min, corresponding modifications have been made in the manuscript.
3. In the manuscript, several cross-reference links to figures and tables are broken.
Response: The Figures and Tables in the manuscript have been re-referenced.
5. The numbering of several figures is incorrect.
Response: Corresponding modifications have been made in the manuscript.
6. References do not follow the MDPI style.
Response: References have been modified.
Technical comments:
Several results reported in this manuscript coincide with the findings of quite old works, such as that of Poppmeier et al., 1966.
Poppmeier, W., Tarmann, B., & Schaaber, O. (1966). Application of alternating electromagnetic fields in the continuous casting of steel. JOM Journal of the Minerals Metals and Materials Society, 18(10), 1109–1114.
Moreover, similar results were already reported in previous works of the authors' research group. Therefore, the authors must state precisely the contributions of this work to state-of-the-art.
The Conclusion and the Article Highlights sections are practically the same. Thus, the authors must unify these two sections in the Conclusions section.
Response: Pulsed magneto-oscillation (PMO) is a new developed and effective homogenization technique for continuous casting. For the industrial application of PMO, its influence on the solidification homogenization of round billet and its stability are worth studying. We outcome of an industrial, full-scale experiment, and the results show that PMO can stably increase the equiaxed grain zone and reduce the macrosegregation of billets. Moreover, it has strong adaptability to steel grade and continuous casting process parameters.
We clearly described the specific contribution of this work in the revised manuscript. Please see in the manuscript.
Reviewer 3 Report
Title: Effects of pulsed magneto-oscillation on the homogeneity of low carbon alloy steel continuous casting round billet
Authors: Ying-chun Wang, Rong-jun Xu, Hong-gang Zhong, Guo-dong Xu, Zhi-shuai Xu, Ren-xing Li and Qi-jie Zhai
General Statement
The paper deals with problem of homogeneity improvement of a metal cast by applying magnetic excitation of internal fluid metal phase convection. With no doubt it fits the Metals journal scope. In addition to the intrinsic credibility value of the results presented, the nature of the experiment is particularly noteworthy: the authors describe procedure and outcomes of an industrial, impressive full scale experiment. In my opinion information of the described studies are worth to be presented and deserve to be shared between the scientific community.
In general, I also found the paper well organized and well written. My reservations are mainly limited to some technical editing errors and a few minor substantive issues. However, there is also one issue that needs discussion. The details are presented bellow. Therefore, I recommend the paper for publication after major revision.
Detailed Comment
My major concern is about the illustration of the PMO effect on liquid temperature field in Fig. 8. I understand that the illustration serves as a schematic and it wasn’t generated in a course of any numerical simulation but according to it the solid wall of the escaped billet is melted again (at the bottom). What mechanism is responsible for that?
Selected Detailed Comments
Page 2, 1-st line from the bottom
The strand casting speeds unit should be corrected.
Table 3
The unit of (Kf Hz) is not clear.
General
I encountered numerous editing errors in my copy. I have marked the locations of the mistakes in the attached file.

Author Response
Dear Reviewer,
Thank you for your kindly comments concerning our manuscript. Those comments are all valuable and very helpful. We have made revisions according to the comments. Revised portion are in red in the revised manuscript. We are outlining the major changes in this letter.
Responses to the comments
My major concern is about the illustration of the PMO effect on liquid temperature field in Fig. 8. I understand that the illustration serves as a schematic and it wasn’t generated in a course of any numerical simulation but according to it the solid wall of the escaped billet is melted again (at the bottom). What mechanism is responsible for that?
Response: Under the action of PMO, double circulation will be generated in the melt. This phenomenon has been confirmed in numerical simulation and experiment, as detailed in references [17,19]. According to the phenomenon of convective heat transfer, it can be find that the liquid temperature at the solidification front of the upper circulation will increase, while that in the lower circulation area is the opposite. This will cause a change in the dendrite growth rate, that is, slow down first and then accelerate. According to the solidification theory, the sudden change of dendrite growth rate and the oscillation of PMO will increase the instability of dendrite and form more dendritic fragments.
Selected Detailed Comments
Page 2, 1-st line from the bottom
The strand casting speeds unit should be corrected.
Table 3
The unit of (Kf Hz) is not clear.
General
I encountered numerous editing errors in my copy. I have marked the locations of the mistakes in the attached file.
Response: The unit of casting speeds has been modified to m/min. And the Figures and Tables in the manuscript have been re-referenced.
Round 2
Reviewer 2 Report
The general comments were correctly addressed. However, there are typos in References [16] and [17].
Concerning technical comments, the new Abstract better reflects the content of the work. Nevertheless, please extend the discussion regarding the differences between the present work and the previous publications of the authors' working group (references [14], [16], [17], and [20]).
Author Response
We have revised the abstract to express the difference between the new job and the previous publications of our group. We hope it can be much clearer.
The revised abstract:
Pulsed magneto-oscillation (PMO) is a new developed and effective homogenization technique and has been successfully applied in rectangular continuous casting, but its processing parameters and effect stability in round billet continuous casting have not been investigated. In this paper, the effects of PMO on solidification structure and macro segregation of Φ 178 mm continuous casting round billets for low carbon alloy steel were studied by industrial experiments. The results show that PMO can stably increase the equiaxed grain area and reduce the macrosegregation of billets. Moreover, it has strong adaptability to steel grade and continuous casting process parameters. Comparing with the billets without PMO treatment, for 93.8% billets (15 billets) solidified with PMO, the equiaxed grain area ratio increases by 5.8% averagely, while for 87.5% billets (14 billets), the carbon segregation index range decreases by 0.06 averagely, though different steel grades, superheat and casting speed were used in the experiment. It’s believed that convection caused by Lorentz force can accelerate the heat dissipation of steel liquid, reduce the temperature of a liquid at the solidification front, while the magnetic oscillation effect is conducive to dendrite fragmentation. Both effects lead to the refinement of solidification structure and reducing of macrosegregation.
Reviewer 3 Report
Title: Effects of pulsed magneto-oscillation on the homogeneity of low carbon alloy steel continuous casting round billet
Authors: Ying-chun Wang, Rong-jun Xu, Hong-gang Zhong, Guo-dong Xu, Zhi-shuai Xu, Ren-xing Li and Qi-jie Zhai
General Statement
While responding to my comments the authors wrote: "Under the action of PMO, double circulation will be generated in the melt. This phenomenon has been confirmed in numerical simulation and experiment, as detailed in references [17,19]. ...". The problem is that my reservation concern not the core of the above mentioned phenomena but only correctness of illustration of them provided in Figure 8. According to it the solid wall of a billet becomes thinner then already was after passing the PMO coil influence zone. This is inconsistent with illustration visible in Figure 1 suggesting monotonously grooving billet wall thickness along pulling direction. In Refs [17] and [19] there is no any indication for this. In my personal experience, the rate of wall thickness increase may change, but it is unlikely that the material will re-melt. Reducing the wall thickness would just mean repeated melting. While accepting all the changes made I ask the authors to consider this and possibly revise the diagram if they feel it is appropriate or explain the phenomenon more fully in the text of the paper.
Summarising my recommendation is: accept after minor revision.
Author Response
Thank you very much for your correction.
Our schematic diagram may indeed cause misunderstanding, so it has been modified. Our original intention is that the forced flow induced by PMO will cause the growth rate of solid shell to slow down first and then accelerate. The dotted line in Figure 8 refers to that the shell thickness of billet without PMO treatment increases evenly along the pulling direction, while the shell thickness increase rate of PMO treated billet decreases first and then increases, but it will not be remelt. The revised drawing shall be more accurate and clearer.